# Personalized Damage Assessment in Aesthetic Surgery: Current Trends and the Italian Scenario

**DOI:** 10.3390/healthcare13212821

**Published:** 2025-11-06

**Authors:** Federico Amadei, Domenico Tripodi, Claudio Cannistrà, Felice Moccia, Marcello Molle, Mario Faenza, Giuseppe Basile

**Affiliations:** 1Hand Surgery Division, C.O.F. Lanzo Hospital, 22020 Alta Valle Intelvi, Italy; amafede@libero.it; 2UniCamillus-Saint Camillus International University of Health Sciences, 00131 Rome, Italy; domenico.tripodi@unicamillus.org; 3Unity of Plastic Surgery, Department of Surgery C.H.U. Bichat Claude Bernard, 75018 Paris, France; 4Multidisciplinary Department of Medical-Surgical and Dental Specialties, University of Campania “Luigi Vanvitelli”, 80138 Naples, Italy; mario.faenza@unicampania.it; 5Department of Biomedical Sciences and Public Health, Marche Polytechnic University, 60121 Ancona, Italy

**Keywords:** legal medicine, aesthetic surgery, informed consent, obligation of means, obligation of result, personalized damage assessment, medico-legal liability

## Abstract

**Introduction**: Aesthetic surgery addresses subjective desires for morphological enhancement and differs from reconstructive surgery due to its elective, non-therapeutic nature. This distinction introduces complex medico-legal challenges, particularly concerning informed consent, patient expectations, and the legal evaluation of aesthetic damage. **Materials and Methods**: A narrative review was conducted using national legislation, Italian and international clinical guidelines, peer-reviewed literature from PubMed, Scopus, and Web of Science, and Italian Supreme Court rulings. Eight commonly litigated aesthetic procedures were analyzed in terms of clinical indications, public reimbursement criteria, and medico-legal risk. **Results**: Findings revealed significant variability in medico-legal exposure among procedures. Fully elective interventions such as liposuction and breast augmentation carried the highest litigation risk. Common legal claims included inadequate informed consent, poor psychological assessment, and mismatched expectations. The review emphasizes the need for personalized consent processes and comprehensive preoperative evaluations. **Discussion**: Italian case law increasingly adopts a “mixed obligation” model for aesthetic surgery, requiring not only technical skill but also a prognostic and relational evaluation of the intervention. Informed consent must be detailed, individualized, and well-documented, as it holds greater legal weight than in therapeutic procedures. Predictive medico-legal tools such as psychological profiling and structured consent protocols are essential for risk mitigation. **Conclusions**: Modern aesthetic surgery requires a redefined approach to damage assessment that incorporates psychological, relational, and identity factors. In both clinical and surgical practice, an approach tailored to the patient’s psychological profile must be increasingly taken into consideration, both when proposing and carrying out treatments and in medical-legal assessments. A legally and ethically sound practice depends on transparency, documentation, and patient-centered care, especially in the absence of therapeutic indications.

## 1. Introduction

Cosmetic surgery encompasses procedures aimed at correcting morphological imbalances or signs of aging, even in the absence of pathology or functional impairment. These interventions are performed on anatomically normal structures and are driven by the patient’s subjective desire to improve appearance according to their personal will. This decision is often influenced by deep psychological, social, or relational motivations that may not be fully rationalized, requiring multidisciplinary preoperative assessment by the surgeon. In this context, the preoperative phase becomes crucial: it is the moment to assess the patient’s suitability, treatment proportionality, and realistic feasibility of the desired result. Even a technically flawless procedure leads to irreversible alterations to body integrity, including scars visible or perceived that can take on a profound subjective significance [1,2]. From a medico-legal standpoint, aesthetic procedures are characterized by their non-therapeutic nature, establishing an atypical contractual relationship with the patient. Since there is no curative intent, physicians are held to a higher standard of information obligation. Italian jurisprudence confirms that, in aesthetic surgery, the physician’s duty extends beyond the correct execution (obligation of means, which requires a party to act to the best of its ability, with reasonable diligence and skill) to a prognostic evaluation of the likely outcome sometimes resulting in a relative obligation of result based on the assumption that the appropriate use of the best means and techniques is linked to the achievement of satisfactory results)based on the adequacy of patient information [3]. The legal value of informed consent in aesthetic medicine surpasses that of therapeutic contexts. It must be specific, structured, and personalized not only addressing surgical risks but also the realistic possibility of failing to meet the patient’s expectations. Indeed, mismatch between objective results and subjective satisfaction is at the heart of most medico-legal disputes in aesthetic surgery [4]. Therefore, documentation of informed consent and the quality of the preoperative discussion are fundamental in assessing professional liability. Legal analysis focuses not only on the technical act but on the entire decision-making pathway, including the management of expectations and where necessary refusal to perform surgery when the risk–benefit ratio is unfavorable [5,6]; Italian Law 219/2017 on informed consent and patient rights, and Law 24/2017 (Gelli-Bianco) on profession liability, as the primary legislative framework. Italian jurisprudence distinguishes between civil liability (breach of contract or tort under art. 1218 c.c.) and criminal liability (personal injury under art. 582–583 c.p.), clarifying the dual medico-legal framework.

## 2. Materials and Methods

This narrative review aims to analyze the regulatory, clinical, and medico-legal framework of cosmetic surgery in Italy, focusing on the implications for damage assessment and professional liability.

### 2.1. Regulatory Framework

The core regulation stems from the State-Regions Agreement of 22 November 2001, issued under Legislative Decree No. 502/1992 and its amendments. Published in the Official Gazette (21 January 2002), it defines the Essential Levels of Assistance (LEA), establishing that the Italian National Health Service (SSN) covers only reconstructive plastic surgery for congenital malformations, trauma, chronic disabling conditions, or neoplasms.

### 2.2. Comparative Regulatory Frameworks

While the Italian medico-legal system provides a detailed civil framework through Law 24/2017 (Gelli-Bianco) and Law 219/2017 on informed consent, regulatory approaches in other jurisdictions reveal significant contrasts. In the United Kingdom, the General Medical Council (GMC) and the Cosmetic Surgery Guidelines (2016, updated 2022) emphasize the surgeon’s duty to ensure informed and reflective decision-making, including mandatory “cooling-off” periods and psychological screening for high-risk patients. In the United States, the legal framework is more decentralized: state-specific medical boards regulate professional conduct, while litigation largely depends on tort law principles of negligence and informed consent. Malpractice claims are often adjudicated under the “standard of care” doctrine, rather than contractual obligations as in Italy. Within the European Union, member states vary in oversight and documentation requirements, but recent directives (e.g., EU Regulation 2017/745 on medical devices) have increased attention to patient safety, traceability, and transparency in aesthetic medicine. Compared with Italy’s jurisprudential evolution toward a “mixed obligation” model, most EU and Anglo-American systems maintain a clearer separation between technical diligence and patient autonomy, offering valuable insights for harmonizing medico-legal standards in aesthetic surgery.

### 2.3. Scientific and Institutional Sources

This review integrates:SICPRE and AICPE guidelines (Italian Society of Plastic, Reconstructive and Aesthetic Surgery and Italian Association of Aesthetic Plastic Surgeon);International clinical-surgical guidelines (ISAPS, ASPS);Indexed scientific literature from PubMed, Scopus, and Web of Science, using keywords such as cosmetic surgery, aesthetic damage, legal medicine, Italy, professional liability.

### 2.4. Aesthetic Procedures Reviewed

Eight of the most common cosmetic surgery procedures were selected for their medico-legal relevance, litigation rate, and the complexity of their indication (Table A1):Abdominoplasty

Indicated for excess skin and abdominal laxity. Covered by SSN only when associated with conditions like hernias, chronic infections, or post-bariatric changes with functional impairment.

2.Blepharoplasty

Correction of eyelid skin/fat excess. Reimbursed only if hypotrophy causes visual field loss (>60%).

3.Otoplasty

Correction of prominent ears. The only aesthetic procedure reimbursed solely for psychological reasons, limited to patients under 14.

4.Rhinoplasty

Nasal reshaping. Reimbursed only when trauma, congenital defects, or breathing obstruction are documented.

5.Mastopexy/Reduction Mammoplasty

For breast lift or volume reduction. Covered in cases of gigantomastia (>500 g/breast) or post-bariatric ptosis with musculoskeletal symptoms.

6.Breast Augmentation

Not reimbursed unless for dysmorphisms (e.g., amastia, asymmetry, Poland syndrome) or post-mastectomy reconstruction.

7.Rhytidectomy (Facelift)

Performed for facial rejuvenation; entirely elective and not reimbursed.

8.Liposuction

Aspiration of localized fat. No therapeutic purpose. Not covered under LEA and associated with risks like embolism or fluid shifts.

### 2.5. Methodological Focus

Each procedure was assessed for:Clinical indicationInsurance and public coverageMedico-legal risksRecurrent litigation patternsCritical issues in informed consent (Italian Law 219/2017 on informed consent and patient rights, and Law 24/2017 (Gelli-Bianco) on professional liability, as the primary legislative framework.)

The aim was to propose a personalized model of damage assessment one that integrates biological, psychological, functional, and relational dimensions, supporting predictive, ethical, and legally sustainable practice.

### 2.6. Literature Review

Recent academic literature highlights the growing complexity of medico-legal aspects in aesthetic procedures. The focus has shifted toward personalized approaches in both clinical and forensic evaluation of damage. Russo et al. developed a reproducible model for assessing aesthetic damage using expert panels and inter-rater reliability. They demonstrated that aesthetic damage cannot rely solely on morphological evaluation but requires expert consensus and subjective appraisal [7]. Piras et al. emphasized the legal distinction between obligation of means and obligation of result in aesthetic surgery. They underscored the importance of customized, not generic, consent as a legal safeguard [8]. Feola et al. conducted a five-year study of malpractice claims in Rome: over 70% of claims involved aesthetic procedures, and most resulted in surgeon liability. They identified deficiencies in prognostic information and consent documentation as major factors [9]. Vicente-Ruiz and Hontanilla introduced the FATIMA acronym, a tool to screen patients at high risk of dissatisfaction and litigation (Female patient using Antidepressants or Anxiolytics, presenting body Tattoos, suing after undergoing a breast surgery with the use of Implants, Middle-aged and with Access to free legal services) [10]. Lim et al. compared legal frameworks across Italy, the UK, and Australia, finding that Italian law places greater weight on individualized consent and professional qualification [11]. Di Paolo et al. explored the rising legal expectations for perfect outcomes even in minor cosmetic procedures, pointing to the symbolic and identity-related weight patients attach to aesthetic surgery [12]. Margara et al. surveyed Italian aesthetic practitioners, revealing low ethical awareness regarding consent, overuse of generic forms, and social media influences [13].

To Clarify: The ‘obligation of means’ implies due diligence without a guarantee of success, whereas a ‘relative obligation of result’ applies when the physician undertakes to achieve a predictable aesthetic improvement under informed conditions. The term ‘mixed obligation’ refers to this intermediate model in aesthetic surgery.

These studies support a multidimensional medico-legal model, combining:Psychological screeningRelational assessmentRisk predictionEnhanced consent documentation

They align with the personalized framework proposed in this paper.

## 3. Results

The analysis of the eight aesthetic procedures revealed a heterogeneous medico-legal landscape, with significant differences in risk exposure depending on the nature of the procedure, its indication, and the adequacy of patient information.

### 3.1. Highest Litigation Risk

Procedures with the highest medico-legal vulnerability were those with:Purely aesthetic indications, i.e., no functional pathologySubjective goals, such as beautification or rejuvenationElective nature, without public reimbursement or therapeutic justification

Notably, liposuction and breast augmentation emerged as the most litigated, due to:Unrealistic patient expectationsLack of psychological screeningPoorly documented or overly generic consentAdverse outcomes with visible scarring, asymmetries, or contour deformities

### 3.2. Common Legal Claims

Across procedures, the most frequently reported medico-legal issues were:Inadequate informed consent (Italian Law 219/2017 on informed consent and patient rights, and Law 24/2017 (Gelli-Bianco) on professional liability, as the primary legislative framework), especially failure to:
Explain possible scarringDescribe likelihood of dissatisfactionWarn of post-op asymmetry or revision surgeryPoor preoperative psychological assessment, particularly in:
Body dysmorphic patientsPatients with depression or external motivations (e.g., partner pressure)Discrepancy between expected and actual results, even when the surgery was technically correct

### 3.3. Factors That Reduced Risk

Litigation risk was significantly reduced when:Consent was personalized, written and explained verballyPsychological profiles were assessed using validated toolsExpectations were aligned with realistic outcomesComplications were explicitly discussed and documentedThe surgeon refused treatment when the risk–benefit balance was clearly unfavorable

## 4. Discussion

Cosmetic surgery, while addressing morphological alterations that may present objective characteristics, is essentially based on the patient’s subjective request, often characterized by a perception of psychological and physical inadequacy [14]. In this context, the medico-legal assessment of any post-operative damage cannot ignore the patient’s initial condition, which is often already marked by a compromised psychological, relational, or identity integrity [15]. Consequently, it is necessary to apply a differential damage criterion, that is, to assess the extent to which the procedure has actually caused a deterioration compared to the pre-existing condition, and not simply a failure to deliver the “promise of beauty” or an aesthetically unsatisfactory outcome. A consolidated legal principle recognizes that surgical complications are statistically predictable, but not always avoidable, events. They do not automatically fall under the professional’s liability, provided there has been valid and documented informed consent, which clearly understands the very possibility of an unfavorable outcome. Liability, therefore, is not based on the event itself, but rather on failure or inadequate management of preoperative information. From a regulatory perspective, the distinction between a duty of means and a duty of result represents a key interpretative pillar. The duty of means, applicable to all medical procedures, implies that the professional must act according to the criteria of diligence, prudence, and skill, without being bound to achieve a specific result. However, in cosmetic surgery due to the lack of therapeutic purposes, there has emerged over time a tendency, including in case law, to assign the surgeon an additional obligation, which can be classified as a “mixed” or “relative duty of result”. This obligation should not be understood in an absolute sense (i.e., as a guarantee of the desired aesthetic outcome), but rather as a commitment to realistically inform on the possibilities for improvement, the failure rate, and the relevance of the re-quested treatment to the initial condition [16]. Numerous rulings by the Italian Supreme Court have addressed this issue, clarifying that in cosmetic surgery, even when the procedure is technically correct, liability may arise due to informational deficiencies, particularly regarding scarring (Cass. Civ. No. 9705/1997), risk–benefit proportionality (Cass. Civ. No. 4677/1998), or the realistic possibility of improvement (Cass. Civ. No. 2409/1989). In other words, failure to achieve the desired aesthetic result does not in itself constitute a physician’s fault but may contribute to a breach of contract if the information provided was omissive, vague, or suggestive. In-formation in cosmetic surgery must be comprehensive, specific, and personalized, including not only surgical and anesthesiologic risks, but also the implicit uncertainty about the outcome and the subjective limits of satisfaction. The surgeon is called upon to perform an educational role, capable of containing unrealistic expectations, promoting an informed decision, and assessing the potential futility or inappropriateness of the procedure. Failure to provide such dissuasion, where appropriate, may constitute professional negligence due to incorrect prognostic assessment, even in the absence of technical error. Significant in this regard is the case law (Cass. Civ. No. 22327/2007), which reiterates the cosmetic surgeon’s obligation to provide detailed information to the patient regarding the objectives, techniques, complications, alternatives, and most importantly the actual foreseeable effects of the procedure. In cosmetic surgery, in fact, the legitimacy of the medical procedure depends even more strictly on the validity and traceability of consent, which should not only be written, but also verbalized, signed in the presence of witnesses, or, if possible, videotaped, in order to document its completeness and effective understanding by the patient. Even contemporary medical-legal doctrine recognizes that in cosmetic surgery, the professional obligation includes a duty to “verify the reasonableness of the request”, that is, to evaluate the balance between aesthetic desire, clinical suitability, and medical relational utility. Otherwise, the procedure, even if performed correctly, may be medically inappropriate and legally unjustifiable. The doctor–patient relationship in cosmetic surgery has a unique relational structure, dominated by emotional, symbolic, and identity-related components [17]. The preliminary phase, in which the therapeutic alliance is built, often lays the foundation for post-operative dissatisfaction and legal litigation [18].

Italian jurisprudence distinguishes between civil liability (breach of contract or tort under art. 1218 c.c.) and criminal liability (personal in-jury under art. 582–583 c.p.), clarifying the dual medico-legal frame-work. Italian Law 219/2017 on informed consent and patient rights, and Law 24/2017 (Gelli-Bianco) on professional liability, as the primary legislative frame-work.

### Medico-Legal Implications

Cosmetic surgery, as a non-therapeutic healthcare service based on a subjective desire for morphological improvement, represents one of the most sensitive and controversial areas of medico-legal liability [19]. Its uniqueness lies not only in the elective nature of the procedure, but also in the fact that the purpose of the service is not the restoration of a compromised function, but rather the adherence to an aesthetic ideal, often individual, that cannot be measured according to objective parameters. In this context, the centrality of the in-formation process and informed consent takes on a preeminent legal and ethical value. A common misconception, even in clinical practice, is that information and consent are overlapping concepts. In reality, they are distinct and complementary aspects of a shared decision-making process. Information constitutes the professional act of the physician, who has the duty to provide the patient with a truthful, comprehensive, and personalized description of the in-formation needed to understand the nature, purpose, limitations, limitations, complications, risks (including statistical risks), and alternatives to the proposed procedure. Information has a unilateral structure, originating from the professional and must be tailored to the individual patient, based on age, education, psychological fragility, cultural level, clinical conditions, and stated or presumed expectations. Consent, on the other hand, represents the patient’s autonomous expression, based on an authentic understanding of the information received. It is a volitional and legally relevant act, whose content must derive from a sufficiently mature cognitive process, allowing the subject to evaluate risks and benefits in relation to their own scale of values. Consent must be personal, free, current, specific, informed, and revocable, as also established by constitutional jurisprudence and the Oviedo Convention (Article 5 et seq.). It follows that information is a “sine qua non”of consent, but it does not exhaust it: the former is objective in nature, the latter subjective. Not every formally expressed consent is legally valid. Consent is lawful only if preceded by adequate and comprehensible information (Cass. Civ., Sez. III, 23 October 2007, no. 22327). In cosmetic surgery, this distinction becomes crucial. A mere signature on a standardized form does not in itself constitute a conscious expression of will, especially when the information provided has not been contextualized and discussed in detail. In Italian law, in-formed consent constitutes the lawfulness of a medical procedure. In the absence of valid consent, the procedure constitutes a criminally relevant personal injury (Article 582 of the Criminal Code), except in cases of necessity (Article 54 of the Criminal Code). In cosmetic surgery, where the clinical indication does not strictly speaking exist, this principle becomes even more compelling. Case law, starting with the Cass. Civ. 9705/1997, 2409/1989, and especially 22327/2007, has consistently reaffirmed that the surgeon may be liable even in the absence of a technical error, if the pre-operative information is found to be flawed, incomplete, evasive, or excessively reassuring. In such cases, a breach of contract exists pursuant to art. 1218 of the Italian Civil Code, with the right to compensation for not only biological, but also moral and existential damages. Information must be active, dialogic, and iterative. One-way or documentary communication, lacking real verbal interaction and verification of understanding, has no legal value. Civil liability also arises in cases where the procedure did not cause physical harm, but failed to meet the patient’s legitimate aesthetic expectations, based on erroneous or ambiguous indications from the doctor. The distinctive feature of cosmetic surgery lies in its non-therapeutic nature, that is, the lack of a curative purpose. In this context, consent not only legitimizes a medical procedure, but replaces the clinical indication, serving as the exclusive criterion of legality. In this sense, the most advanced medical-legal doctrine [20,21] has proposed a distinction between therapeutic consent (based on a shared clinical indication) and aesthetic consent (based on a subjective assessment of the benefit and a realistic representation of the result). In the latter case, the surgeon must perform a prognostic-relational assessment, that is, inform not only what he or she can technically do, but also what makes sense to do, taking into account the patient’s psyche, socio-relational context, and expectations. Aesthetic information must include the degree of likelihood of the expected improvement, the possibility that the result will be perceived as inadequate, and the concrete relational utility of the procedure [22]. An emerging proposal in the literature is the application of a predictive forensic medicine model to cosmetic surgery, including preoperative psychological profiling (including validated tools such as PSE-Q or BAT), enhanced documentation (customized multilingual forms, video recordings of consent), and medicolegal risk scales based on variables such as the location of the procedure, expressed expectations, risk of complications, or visible scarring. In this framework, informed consent is no longer simply a formal act, but a structured, verifiable, and measurable procedure. The medicolegal training of cosmetic surgeons should include knowledge of clinical communication, bioethics, health law, civil liability, and forensic writing techniques. In cosmetic surgery, in-formed consent therefore represents not only a legal protection tool but also the central act of ethical and professional responsibility. Informing means guiding the patient to be aware of their limitations, to scale back their expectations, and to assess risks, including non-technical ones [23]. Only with genuine, comprehensive, dialogue, and documented consent can a healthcare contract in cosmetic surgery be considered validly established, failing which strict liability arises for unnecessary, unapproved, or unjustifiable procedures. The future of cosmetic surgery depends on a renewed medical-legal alliance, founded on transparency, ethics, personalization, and documentable decision-making [24].

## 5. Conclusions

Quantifying damage in cosmetic surgery requires a sophisticated medico-legal approach that goes beyond the mere assessment of psychophysical integrity to embrace the complexity of the patient’s bodily and relational identity. Aesthetic damage, traditionally considered a subcategory of biological damage, now takes on an independent meaning, as it affects not only the morphological dimension but also the communicative, expressive, and identity spheres of the individual. The Constitutional Court, with ruling no. 184 of 1986, recognized the significant nature of damage to physical appearance as a violation of a constitutionally protected right. More recent legal doctrine and case law have broadened its scope, considering aesthetic damage as a prejudice that can compromise facial expression and physiognomy, and therefore the individual’s social and professional projection. In criminal law, damage to the aesthetic integrity of the face can constitute a permanent disfigurement relevant under Article 583 of the Criminal Code, with legal implications even in the absence of functional impairment. In civil law, case law has clarified that aesthetic damage must be assessed individually, taking into account the physical, psychological, relational, and work-related characteristics of the injured party, and in relation to the concrete repercussions on their daily life. Adverse aesthetic outcomes can lead to reduced relational capacity, a loss of professional opportunities, or future financial damage, thus constituting a specific impairment of working capacity, distinct from moral or biological damage strictly speaking. The medico-legal assessment of aesthetic damage, therefore, cannot be standardized but must use flexible clinical-forensic tools capable of integrating morphological, functional, psychosocial, and existential dimensions. It is necessary to analyze the patient’s individual profile, pre-existing condition, expectations, potential repercussions on their emotional, work, and social life, and the possibility of even partial aesthetic recovery. This personalized approach is particularly essential in cosmetic surgery, where the demand for services is motivated by subjective, not therapeutic, needs, and where dissatisfaction can also arise from a technically correct result that is psychologically unacceptable to the patient. Ultimately, the evolving concept of damage in cosmetic surgery requires forensic doctors to perform a complex and multidimensional assessment, based not only on anatomical criteria but also on identity, communication, and relational factors. To be ethically and legally sustainable, modern cosmetic surgery must embrace a new culture of damage: one centered on the person, on the subjectivity of the experience of harm, and on the full documentation of the information and decision-making process that preceded the procedure.

### Recommendations

Surgeons should integrate psychological screening (e.g., validated tools like PSE-Q or BAT), structured consent templates with video or witness documentation, and maintain the ethical right to refuse surgery when expectations are unrealistic or disproportionate.

## Data Availability

No new data were created or analyzed in this study.

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
