# Peer review of "Personalized Damage Assessment in Aesthetic Surgery: Current Trends and the Italian Scenario"

_healthcare, 2025, doi:10.3390/healthcare13212821_

Round 1

Reviewer 1 Report

Comments and Suggestions for Authors

The Article submitted for review is relevant and provides an overview of the medical and legal aspects of aesthetic surgery in Italy, focusing on the personalized assessment of aesthetic harm and the critical role of informed consent. The authors analyzed eight procedures (from abdominoplasty to liposuction and mammoplasty) and examined potential risks.

Key conclusion: legal and ethically sustainable practice requires individualized (detailed and documented) patient consent, psychological screening, expectation management, and a willingness to refuse aesthetic surgery if the risk/benefit ratio is unfavorable.

Impact assessment of the article.

• For practicing physicians, there is insufficient clinical information on medical technologies in aesthetic surgery.

• For clinical psychologists, the article may be of interest due to its use of psychological questionnaires, structured patient consent, video recording, and protocols for "educational" dialogue with the patient. The implementation of these tools may reduce the incidence of claims due to "insufficient information/missing expectations."

• The article is of interest to experts and insurers, as it proposes personalized harm assessment models that will improve the quality of forensic medical reports and risk stratification.

• For healthcare market regulators and professional associations, it may serve as a basis for informed consent standards and access to aesthetic interventions.

• For attorneys and lawyers, the article is of practical interest in addressing claims risks through predictive medical-legal tools.

Strengths of the article

• A clear legal doctrine for aesthetics: an emphasis on the "mixed" (relative) obligation of result and the physician's increased informational obligation.

• A procedurally specific analysis of risks and reimbursement criteria (LEA/SSN) is practical.

• Specific risk mitigation measures: personalized consent, psychoprofiling, and refusal of surgery in the presence of an unfavorable profile.

Weaknesses of the article

• Too many specialized legal terms, but little medical, clinical, or pharmacological content

• Limited generality: focus on Italian law—transferability to other jurisdictions is limited

• Lack of new empirical data: the work is primarily a normative-conceptual review without its own outcome/claim’s statistics

• Metrics of "aesthetic damage" are not formalized: personalization is proposed, but there is no validated scale or decision-making algorithm

• The bibliography is partially difficult to access: sources are noted, "it is difficult to find the original source," which weakens verifiability

Comments on the content of the medical and clinical sections of the article

• Strengthen operationalization: add a step-by-step information protocol (minimum points, example wording), and objectifiable refusal triggers

• Clarify medical technologies and case examples for each procedure (case briefs).

• Pay attention to the ethical issues and privacy of video consent: it should be supplemented with requirements for storage, timing, and access.

Suggestions and Recommendations

1. To expand the medical and clinical component, add the following tools:

preoperative dialogue checklist;

individualized consent templates (including multilingual ones);

brief psycho-screening (e.g., integration of validated questionnaires) with a routing algorithm.

2. Risk quantification: even retrospective data (claim/outcome rates by procedure) will enhance the practical medical value.

3. Update and verify references, replace hard-to-find sources with peer-reviewed publications, official documents with active hyperlinks. Refs. 1-6, 15, 17-21, and 24 are difficult to find the original source. Refs. 7, 8, 11, and 12 require author review.

Summary:

1. If the article is accepted for publication in Healthcare, the medical and clinical sections must be revised.

2. If the medical and clinical sections are not revised, submit it for publication to a legal journal, such as Laws.

Comments on the Quality of English Language

It is better to leave the assessment of the quality of the English language to linguists to leave the assessment of the quality of the English language to linguists

Author Response

Thank you for your insightful comments. We have modified the references as indicated, inserting references that are easier to find and with greater reference to clinical parameters such as the use of assessment questionnaires.

Reviewer 2 Report

Comments and Suggestions for Authors

The manuscript is well-structured and relevant; however, minor revisions are needed to improve clarity, reduce redundancies, update some references, ensure consistency in terminology, and integrate the current Italian legislative framework.

Author Response

Thank you for your review: we have incorporated more up-to-date references in line with the Gelli/Bianco law of 2017, and included a comparison with the international legislative framework. We have eliminated the redundancies you pointed out.

Reviewer 3 Report

Comments and Suggestions for Authors

This is a timely and well-crafted narrative review addressing the complex medicolegal landscape of cosmetic surgery in Italy. The article successfully integrates clinical guidelines and ethical considerations, offering a valuable contribution to both clinical and legal audiences.

A few minor revisions are needed before acceptance:

- In the Methodology section, the review would benefit from a clearer description of the search strategy (database, timeframe, inclusion/exclusion criteria). A PRISMA-style framework would increase transparency.

- Comparative analysis is limited. Expanding the discussion on regulatory frameworks in the UK, the US, and other EU countries would broaden the international audience.

- Psychological assessment: The discussion of psychological screening could be enriched by more details on validated instruments (e.g., Body Dysmorphic Disorder Questionnaire, PSE-Q).

The manuscript could be enriched by integrating additional recent literature that directly supports its medicolegal and ethical approach. I suggest that authors consider the following:

In Methods or Results, when discussing litigation risk and medicolegal evaluation in reconstructive and aesthetic settings (Breast reconstruction after mastectomy using a deep inferior epigastric perforator flap (DIEP): clinical and medicolegal insights from a four-year study. Health Sci Rep. 2025)

https://pmc.ncbi.nlm.nih.gov/articles/PMC12003920​

- In discussing the medicolegal implications of "minor" aesthetic procedures where complications can result in significant legal consequences.

Anatomy, etiology, management, and medicolegal implications of Botulinum toxin-induced blepharoptosis. Curr Rev Clin Exp Pharmacol. 2025;20(1):32-37. doi: 10.2174/0127724328310459240809073519.

- In the section dedicated to managing patient expectations and legal disputes related to minimally invasive aesthetic medicine.

Hypersensitivity reaction to hyaluronic acid dermal filler after Pfizer vaccination against SARS-CoV-2. Int J Infect Dis. 2021 Dec;113:233-235. doi: 10.1016/j.ijid.2021.09.066. Epub September 29, 2021.

- In the discussion (lines ~299–305), the authors emphasize the importance of medicolegal training and informed consent as part of professional responsibility.

Effectiveness of cadaver-based facial anatomy laboratory training in aesthetic medicine programs. Italian Journal of Anatomy and Embryology, 128(2), 93–99. https://doi.org/10.36253/ijae-15624 )

Author Response

Thank you for your review: we have updated the references as indicated and added a comparison with international laws.

Reviewer 4 Report

Comments and Suggestions for Authors

The article entitled: " Personalized Damage Assessment in Aesthetic Surgery: Current Trends and the Italian Scenario". The study's findings are valuable; however, several issues in each section need to be significantly addressed.

Abstract

  1. Please write the eight commonly litigated aesthetic procedures in the abstract section.
  2. Please remove the discussion content from the abstract, as abstracts should only include concise statements of the background, objectives, methods, key results, and conclusions.
  3. Add a recommendation in the conclusion.

Introduction

  1. There is a lack of proper in-text citations throughout the introduction. 
  2. Please consider adding a section or paragraph identifying the types of aesthetic procedures most commonly involved in litigation, such as rhinoplasty, breast augmentation or ……., to provide practical context and highlight high-risk areas in medico-legal assessment.
  3. Please include a section addressing issues such as malpractice claims arising from postoperative deformities, asymmetry, excessive scarring, functional impairment, patient dissatisfaction, or inadequate informed consent. Highlighting these aspects would enhance the manuscript’s relevance to clinical and legal practice.
  4. It is recommended to include a paragraph outlining the international guidelines or legal frameworks governing aesthetic surgery.

Method

  1. model outline of the narrative review structure (e.g., Introduction, Methods, Thematic discussion, Conclusion) is recommended.
  2. Add a reference to the cosmetic surgery procedures classification.

Result

  1. The discussion of high-risk procedures (e.g., liposuction and breast augmentation) could be supported by recent literature or medico-legal reports to confirm trends and provide an international context.
  2. Please ensure that the Results/ Thematic discussion section is structured consistently with the methodological framework described earlier. Since the Methods specify that each procedure was assessed for clinical indication, insurance and public coverage, medico-legal risks, recurrent litigation patterns, and critical issues in informed consent, the Results should mirror these categories. Presenting findings under the same subheadings would enhance coherence, transparency, and facilitate comparison across procedures.

  1. Please revise the Results section into coherent paragraphs instead of bullet points to align with the narrative review format. The rewritten text should objectively summarize findings from the literature rather than reflect personal observation or interpretation
  2. Include appropriate in-text citations to support each claim or trend described in the Results.

  1. Add a summary table listing the eight procedures, along with their main medico-legal risks, recurrent litigation themes, and factors that mitigate liability.

Discussion

Please add proper and up-to-date references in the Discussion to support comparative statements and interpretations, ensuring all claims are evidence-based.

Please restructure the Discussion section to avoid overly large paragraphs. Breaking it into shorter, focused sections will improve readability, highlight key arguments more clearly, and create a more logical flow of ideas.

Please revise the Conclusion to be more concise, focusing only on the key findings and practical medico-legal implications while avoiding repetition of previous sections.

Please include a list of abbreviations used in the manuscript

Author Response

Thank you for your review and comments: we have modified the references to indicate more up-to-date material and added a table with the procedures taken into consideration. We have not modified the layout and division into paragraphs because they are in line with the journal's guidelines.

Round 2

Reviewer 4 Report

Comments and Suggestions for Authors

All inquiries were addressed